# [^18^F]FDG PET/CT in Large Vessel Vasculitis: The Impact of Expertise and Confounders on Image Analysis

**DOI:** 10.3390/diagnostics12112717

**Published:** 2022-11-07

**Authors:** Lidija Antunovic, Alessia Artesani, Michael Coniglio, Wim J. G. Oyen, Michele Ciccarelli, Carlo Selmi, Arturo Chiti, Martina Sollini

**Affiliations:** 1IRCCS Humanitas Research Hospital, Via Manzoni 56, Rozzano, 20089 Milan, Italy; 2Department of Biomedical Sciences, Humanitas University, Via Rita Levi Montalcini 4, Pieve Emanuele, 20090 Milan, Italy; 3Department of Radiology and Nuclear Medicine, Rijnstate Hospital, 6815 AD Arnhem, The Netherlands

**Keywords:** vasculitis, [^18^F]FDG PET/CT, Takayasu arteritis, giant cells arteritis

## Abstract

Background: Diagnosis of vasculitis is challenging. To avoid invasive approaches, clinical guidelines recommend the use of diagnostic imaging. This study aimed at evaluating the diagnostic accuracy of [^18^F]-fluorodeoxyglucose ([^18^F]FDG) position emission tomography/computed tomography (PET/CT) in large vessel vasculitis (LVV) and how this is affected by inter-operator variability. Methods: A total of 279 patients who performed [^18^F]-FDG PET/CT for suspicion of LVV were retrospectively analyzed. We tested the qualitative and semi-quantitative analysis and parameters influencing image quality and interpretation. Exams were evaluated by two readers with different experience and their performance was compared. Results: LVV diagnosis was confirmed in 81 patients. [^18^F]-FDG PET/CT accuracy was 73% and 67% for the expert reader and less experienced reader, respectively. The expert reader overall performed better than the less experienced one, with higher accuracy in patients with normal BMI (77.3 vs. 63.8%), normal level of glycemia (73.3 vs. 65%), younger age (76.6 vs. 68.2%), and when no therapy was in course at time of imaging (76.7 vs. 66.7%). The diagnostic performance of both readers did not improve using semi-quantitative parameters. Conclusions: We confirmed the appropriateness of the recommended criteria for image acquisition and interpretation, underlining the importance of experience in image interpretation for the optimal diagnostic performance of [^18^F]FDG PET/CT in vasculitis.

## 1. Introduction

Systemic vasculitis encompasses multiple subtypes of chronic inflammatory diseases affecting blood vessels of any size (artery, arteriole, vein, or capillary) [1] with widely variable clinical manifestations. Large vessel vasculitis (LVV) include Takayasu arteritis and giant cells arteritis (GCA), characterized by inflammation of the major aortic branches or aorta itself [2], with the second being more frequent in women, with incidence rates between 1.6 and 32.8 cases/100,000 persons over 50 years of age [3].

Takayasu arteritis recognizes an early phase with non-specific symptoms such as fever, fatigue, or weight loss, and a later phase following the involvement of the carotid arteries and aorta, causing carotidynia, neck pain, and back pain, resulting in decreased or absent peripheral pulses. GCA manifests with abrupt headaches associated with other less frequent symptoms such as impaired vision and claudication of the masseter muscles [4,5].

Despite the new American College of Rheumatology classification criteria [6], the diagnosis of LVV remains challenging and still requires arterial biopsies in most cases, despite the fact that this may be frequently spared. The 2018 European League Against Rheumatism (EULAR) recommendations for the clinical use of different imaging modalities for the diagnosis of vasculitis arrayed the available evidence and pointed at imaging techniques to overcome the need for invasive measures [7].

[^18^F]-Fluorodeoxyglucose ([^18^F]FDG) position emission tomography/computed tomography (PET/CT) has been proposed for the diagnosis and follow up of LVV [8] as it also detects LVV-associated inflammation with good sensitivity and specificity at the early stages of the disease [9,10]. In 2018, the joint procedural recommendations by the European Association of Nuclear Medicine (EANM), Society of Nuclear Medicine and Molecular Imaging (SNMMI), and the PET Interest Group for [^18^F]FDG PET/CT imaging in LVV aimed at standardizing all procedural steps, from patient preparation to image acquisition and reporting [11]. Visual assessment remains largely operator dependent and requires a significant experience in the field. Although there is no standard method for visual interpretation in LVV, use of the grading system is suggested to compare the vascular to the liver uptake. Methods of semi-quantitative image assessment have also been proposed based on the measurement of the standardized uptake values (SUV) of the arterial wall compared to a reference organ (e.g., liver, venous wall, blood pool) [12,13]. Nevertheless, there is currently no clear consensus on the interpretation criteria for PET imaging in vasculitis.

We retrospectively evaluated the diagnostic accuracy of [^18^F]FDG PET/CT in suspected LVV using different image analysis approaches (i.e., qualitative and semi-quantitative) and testing different parameters that may influence the exam quality and image interpretation. Finally, we compared the performance of the readers with different experience to account for inter-operator variability.

## 2. Materials and Methods

### 2.1. Study Design

In this observational study, we retrospectively screened all patients who underwent a PET/CT with [^18^F]FDG for a suspected or established diagnosis of vasculitis in IRCCS Humanitas Research Hospital from January 2012 to December 2021. Subsequently, only patients with available PET/CT images and clinical follow-up were included in the study. Initially, 432 patients were selected by screening the medical reports, using the keywords “vasculite”, “vasculitico”, or “arterite”. Clinical diagnosis was used as the reference standard. This was established based on the clinical presentation, laboratory tests, and imaging. More complex and inconclusive cases were handled by interdisciplinary discussion, and final diagnosis was established accordingly, with regard to good clinical practice at our Institution. One hundred and fifty-three patients were excluded, as these were external referrals for PET/CT with no follow-up data in our institution’s database. Overall, 279 patients were included in the analysis. The demographic characteristics of patients (i.e., gender, age, weight, height, body-mass index (BMI)), clinical and laboratory data (i.e., type of vasculitis, if ongoing corticosteroid treatment, erythrocyte sedimentation rate (ESR), and concentration of c-reactive protein (CRP)) were retrieved from the electronic records of the hospital. All information regarding PET/CT imaging including pre-injection blood glucose level, administered [^18^F]-FDG activity, elapsed time between injection and acquisition, and scanner type were also recorded. The study was approved by the Ethics Committee of IRCCS Istituto Clinico Humanitas (approval no. 53/21, date 14 December 2021).

### 2.2. [^18^F]FDG PET/CT Acquisition Protocol

Prior to [^18^F]FDG administration, the glucose levels were checked in fasting patients (at least 6 h) and if lower than 200 mg/dL, intravenous injection of [^18^F]FDG (~6 MBq/kg) was performed. PET/CT images were acquired approximately 60 min after administration following the EANM guidelines [11] using one of two integrated EARL accredited (http://earl.eanm.org/cms/website.php (accessed on 23 October 2022)) PET/CT scanners: a second generation Siemens Biograph LS 6 scanner (Siemens, Munich, Germany) equipped with LSO crystals and a six-slice CT scanner (denominated P1), or a third generation GE Discovery PET/CT 690 equipped with LYSO crystals and a 64-slice CT scanner (General Electric Healthcare, Waukesha, WI, USA) (denominated P2).

### 2.3. Image Analysis

PET/CT images were retrieved from the institutional picture archiving and communication system (PACS) and visually assessed by an experienced nuclear medicine physician (LA, reader 1) with extensive expertise in inflammation imaging reporting (7 years). A student in their final year of medical school acted as the second reader (MC, reader 2).

During visual assessment, both readers independently examined all vascular districts including extracranial arteries (e.g., temporal and vertebral ones). Each PET/CT was defined as positive for vasculitis if the linear circumferential [^18^F]-FDG uptake of at least one vascular region of interest was increased compared to the physiological uptake of the liver [11]. Whenever the vascular target district was represented by femoral arteries, only markedly increased [^18^F]FDG uptake compared to the liver was considered positive. Semi-quantitative analysis of the PET/CT images was also performed by measuring the maximum standardized uptake value (SUV_max_), drawing a region of interest (ROI) on the axial images in 14 different vessels of interest: right and left carotid, subclavian and axillary arteries, ascending aorta, aortic arch, descending aorta, abdominal aorta, right and left iliac, and femoral arteries. Background uptake, measured as SUV_max_ in the liver and in the inferior vena cava (IVC) was calculated by drawing a ROI in the right lobe of the liver and in a region of the venous wall approximately at the medium level of lumbar column for the IVC, respectively. Vascular-to-liver and vascular-to-IVC ratios were calculated for each vessel. The value of these ratios was compared to the cut-off values found in the literature of 1.0 for vascular-to-liver and 1.6 for the vascular-to-IVC ratio [12,13], and separate analysis for liver and IVC was conducted. First, to determine the performance of the semi-quantitative analysis, an exam was classified as positive if the vascular-to-liver and vascular-to-IVC were higher than each cut-off value, respectively, in at least one vessel. Exams were defined as negative if the vessels had a vascular-to-liver and vascular-to-IVC ratio lower than the respective cut-off.

Second, the utility of semi-quantitative analysis over visual analysis was also tested. In this regard, in the case of a PET exam visually assessed as negative but presenting a vascular-to-liver ratio higher than the reference cut-off for liver in any vascular region, the examination was re-classified as positive for LVV in the semi-quantitative analysis. The same switch from a negative to a positive exam was performed when the vascular-to-IVC result was higher than the IVC reference cut-off. Exams visually interpreted as positive by the readers were not re-classified irrespectively of the reference cut-off.

The Xeleris^TM^ workstation (General Electric Healthcare, Waukesha, WI, USA) was used for visual and semi-quantitative analysis.

### 2.4. Statistical Analysis

Statistical analysis was carried out using STATA 17 software [14]. First, we evaluated the diagnostic performance of each reader by comparing their findings with the clinical diagnosis (reference standard). Sensitivity, specificity, positive predictive value (PPV), and negative predictive value (NPV) were calculated. Subsequently, Cohen’s Kappa coefficient was estimated to assess the inter-rater agreement, and the interpretation proposed by Landis and Koch for inter-rater reliability was adopted (Table 1) [15].

Quantitative variables were tested for normal distribution by the Shapiro–Wilk test. To compare the distribution of the values, a non-parametric Mann–Whitney U test was performed.

Sub-analysis to test the diagnostic performance of each reader according to specific criteria was also performed. Specifically, we assessed whether the diagnostic performances might be affected by age (≤65 years versus >65 years), BMI (normal versus overweight), ongoing corticosteroid therapy (yes versus no), levels of pre-injection glycemia (≤ or >126 mg/dL), early, timely, or delayed images (according to timing suggested by EARL, and the time between the radiotracer injection and image acquisition <60 min, between 60 and 65 min, and >65 min, respectively) and scanner (P1 versus P2).

The semi-quantitative analysis was performed first by applying the cut-off to each vascular district, regardless of the visual assessment and comparing it to clinical diagnosis (reference standard). Finally, the added value of semi-quantitative analysis was calculated for each reader, assessing how often an exam initially visually interpreted as negative switched to positive by applying the reference cut-off and using the clinical diagnosis as the reference standard.

## 3. Results

Elderly patients (>65 years) accounted for 172 out of 279 cases (62%). BMI was normal (≤25) in 141/279 patients (51%). One hundred and fifty patients were taking oral corticosteroid therapy at the time of imaging (54%). Pre-injection blood glucose levels were within normal range (≤126 mg/dL) in 240/279 patients (86%). Timely images were acquired in 79 cases. Table 2 summarizes the main characteristics of patients included in the analysis. ESR and CRP were available only in a subgroup of patients (97/279 and 161/279 patients, respectively.) Both ESR and CRP mean values were out of the upper range (49 ± 32 mm/h and 9 ± 12) mg/L.

In total, 81 patients (29%) were clinically diagnosed with LVV. Reader 1 and reader 2 failed diagnosis in 76 patients (37 false negative and 39 false positive) and 92 patients, respectively (33 false negative and 59 false positive). Figure 1 and Figure 2 show examples of a true positive and a false positive PET exam. Among the 76 patients wrongly classified by reader 1, 51 were >65-years-old, 44 were overweight, 46 were taking steroids, 12 presented with hyperglycemia, 29 were early acquired, while in 26 patients, the acquisition was delayed, and 23 were scanned with P1.

Among the 92 patients wrongly classified by reader 2, 58 were >65-years-old, 41 were overweight, 49 were taking steroids, eight presented with hyperglycemia, 30 were early acquired, while in 34 patients, the acquisition was delayed, and 22 were scanned with P1.

Collectively, 22/76 patients wrongly classified by reader 1 and 22/92 patients misclassified by reader 2 were elderly overweight patients who were taking steroids at the time of imaging. Figure 3 shows an example of a false negative PET exam during corticosteroid treatment.

The diagnostic performance of reader 1 and reader 2 are detailed in Table 3 and Table 4, respectively. In the entire population, the sensitivity, specificity, accuracy, PPV, and NPP of reader 1 were 54%, 80%, 73%, 53, and 81%, respectively. The diagnostic performances of reader 2 were sensitivity = 59%, specificity = 70%, accuracy = 67%, PPV = 45%, and NPP = 81%. Figure 4a is a radar chart summarizing the performances for both readers considering the clinical diagnosis of vasculitis.

Reader 1’s best performance was attained in patients with a normal BMI (sensitivity of 65%, specificity of 83%, and 77% accuracy). Seventy out of 81 patients (86%) with a final diagnosis of vasculitis were taking steroids at the time of imaging, and 38 of them (54%) were correctly identified by reader 1. The lowest sensitivity (35%) was observed in patients scanned with P1, although in the same subgroup of patients, reader 1 was so specific (89%) to be even slightly more accurate than in patients imaged using P2 (76% versus 71%, respectively). Reader 2 performed similarly to reader 1, with an accrued decline in performance in the aforementioned subsets of patients (Table 4). Forty-one out of 70 patients (58%) who assumed steroids at the time of imaging and with a final diagnosis of vasculitis were correctly identified by reader 2.

Overall, the interrater agreement for accuracy between the experienced nuclear medicine physician and the student was 71% with a Cohen’s Kappa value of 0.37 (fair agreement). A radar chart summarizing the performances of reader 2 against reader 1 is presented in Figure 4b.

The agreement between the nuclear medicine physician and the student was scarce (from fair to moderate), even when considering sub-analysis (Appendix A). The best agreement was observed in patients ≤65-years-old, with increased blood glucose level, and in those early imaged (Cohen’s Kappa = 0.50, 0.54, and 0.53, respectively).

In all vascular regions, the mean SUV_max_ values were slightly higher in patients with vasculitis, but only those calculated in the axillar arteries were significantly different compared to patients without vasculitis. Mean SUV_max_ values calculated in the liver and IVC were comparable in the two groups (Appendix A).

Using only semi-quantitative analysis, the sensitivity, specificity, accuracy, PPV, and NPV were 21%, 87%, 68%, 41%, and 73%, respectively, when using liver cut-off, while when applying the IVC cut-off, the sensitivity was 85%, specificity 21%, accuracy 39%, PPV 31%, and NPV 74%.

The diagnostic performance of both readers when applying the cut-off values for the liver and IVC, respectively, was reassumed in Appendix A. Compared to the visual analysis, the liver cut-off approach slightly increased the sensitivity similarly (54% versus 59% for reader 1 and 59% versus 62% for reader 2) of both readers and reduced specificity (80% versus 73% for reader 1 and 70% versus 64% for reader 2). Using the IVC cut-off, the sensitivity of both readers markedly increased (54% versus 93% for reader 1 and 59% versus 93% for reader 2), but the specificity drastically dropped (80% versus 17% for reader 1 and 70% versus 15% for reader 2). Adding semi-quantitative analysis to the visual one, the accuracy of both readers was lower when compared to the visual approach alone, regardless of the reference used (liver or IVC).

## 4. Discussion

Our paper retrospectively evaluated the performance of [^18^F]FDG PET/CT in diagnosing vasculitis by comparing the reader ability of an experienced nuclear medicine physician to a medical student. As expected, the experienced nuclear medicine physician outperformed the medical student (accuracy of 73% and 67%, respectively) resulting in higher specificity (80% versus 70%). The literature base underpinning the evidence about the role of [^18^F]FDG PET/CT to identify and monitor LVV is continuously increasing [10,11,16,17], and our findings confirmed that an appropriate learning curve and significant expertise in the field are essential, since many factors may affect image analysis.

First, we considered age as a potential confounder of image interpretation. Although patient demographics were consistent with the expected age and sex distributions for LVV [3], some conditions such as atherosclerosis were reported to affect the vascular [^18^F]FDG signal. We considered age as a surrogate of atherosclerosis and accordingly, we divided and analyzed the population in young (≤65 years) and elderly (>65 years) subjects, although many other factors, conditions, and diseases including cardiovascular ones, might influence, beyond age, the vascular [^18^F]FDG uptake [17,18,19,20]. Diagnostic accuracy was higher in the young than in the aged patients for both readers, suggesting that age may act as a confounder. As aging and BMI may have an influence on vascular [^18^F]FDG uptake [21], overweight patients are more prone to atherosclerosis. Moreover, the quality of images is poorer in overweight patients than in subjects with a normal BMI [22]. As expected, the sensitivity in overweight patients was lower than in normal individuals for both readers, even if the specificity was relatively high (78% for both readers in overweight versus 83% for reader 1 and 63% for reader 2 in normal patients). Steroids are listed among the factors that may affect the diagnostic performance of [^18^F]FDG PET/CT in LVV diagnosis, reducing vascular [^18^F]FDG uptake, and increasing the signal in the liver. Although Nielsen et al. [23] reported a very short diagnostic window (i.e., 3 days) between steroid initiation and [^18^F]FDG PET/CT examination, discontinuation or postponement of steroid therapy could expose patients with GCV or Takayasu arteritis to complications [11], making it difficult to comply with the appropriate timing. As expected, the accuracy of reader 1 was higher in patients not taking steroids, although in our cohort, a high proportion of patients who were diagnosed with vasculitis took steroids at the time of imaging (86%). Reader 1 correctly identified 54% with a final diagnosis of vasculitis patients who took steroids at time of PET/CT, suggesting that the proper and stringent use of diagnostic criteria as recommended by the guidelines [11] might partially overcome this limitation. The accuracy of reader 2 was comparable independent of steroid use, and 41 out of 70 patients who took steroids at the time of PET/CT with a final diagnosis of vasculitis were correctly identified by reader 2. Interestingly, the student labeled more positive patients who were taking corticosteroid therapy than the expert physician, as denoted by the slightly increased sensitivity (64% versus 55%). This may be explained by considering that an inexperienced reader might tend to evaluate them as positive ambiguous cases to avoid missing the diagnosis. It is also known that even a low maintenance dose of steroids is effective in suppressing [^18^F]FDG uptake in the vascular wall [24]. In our study we did not take into consideration the duration of treatment and/or different dosage of the steroid therapy, since these data were not available for all patients due to the retrospective design of the study.

Although pre-injection blood glucose levels seem to be less important in infection and inflammation than in oncology [11,25,26], a negative correlation between fasting glucose level and arterial [^18^F]FDG uptake has been reported [27], supporting a recommended value of pre-injection blood glucose values lower than 126 mg/dL [11]. In our series, glycemia did not impact the performance of the experienced reader (sensitivity of 54% in patients with low as well as in those with high blood glucose levels), but it affected reader 2’s performance, decreasing the sensitivity in the case of lower blood glucose levels (accuracy of 65% versus 79% in patients with low versus high blood glucose levels, respectively). These results seemed to contradict the above-mentioned evidence reported by Bucerius et al. [27], who suggested that in the case of hyperglycemia, analysis should be mathematically corrected for the vascular [^18^F]FDG uptake. However, the study of Bucerius et al. [27] was focused on semi-quantitative parameters and they did not investigate the effect of pre-injection glycemia on visual assessment. Moreover, the distribution of data should be considered. Only 39 patients in our cohort experienced high levels of blood glucose, and only 13 of them had a final diagnosis of LVV, preventing further speculations. The current guidelines [11] recommend at least 60 min between intravenous administration of [^18^F]FDG and acquisition, although it has been reported that delayed images, reducing blood pool activity, resulted in a higher accuracy compared to timely acquisition [28]. This was not corroborated by our data, since we did not observe significant improvement in the performances in the case of delayed images, even when considering timing according to the EARL guidelines [25] (Appendix A). These findings suggest that the use of proper criteria for image interpretation might reduce the impact of factors related to image acquisition. Finally, the scanner might impact the quality of the images and therefore on the diagnostic performance. Both readers were less sensitive in analyzing images acquired using the second generation P1 scanner than those obtained with the third generation P2 scanner. Currently, there are no specific recommendations for acquisition protocols in LVV. At our institution, we acquired LVV patients from vertex to knee (at least), and as a general rule, we preferred to scan heavy patients with P2.

Generally, when comparing the evaluations by both readers, there was a fair interrater agreement, as indicated by Cohen’s kappa. The student’s diagnostic performance was inferior but fair over all parameters compared to the experienced physician’s, except for evaluations made in patients who had their imaging studies performed on the scanner P1, and to a lesser extent in overweight patients.

Collectively, focusing only on patients wrongly classified by reader 1, just under a third were elderly overweight patients who took steroids at the time of imaging. Considering the patients misclassified by reader 2, and comparing them to those of reader 1, other factors such as the scanner seeming to interfere with image interpretation, suggesting that in an inexperienced reader, all elements probably co-occurred, affecting the image analyses.

The diagnostic performance of readers 1 and 2 was not improved by adding semi-quantitative analysis to the visual one. The use of semi-quantitative parameters to diagnose LVV is not recommended unless within the context of clinical trials [11,17]. Indeed, there is no evidence that semi-quantitative [^18^F]-FDG PET/CT metrics may help to better diagnose LVV than visual scoring [17]. Moreover, a reduced diagnostic accuracy of target-to-background based semi-quantitative indices has been reported in patients under glucocorticoids [29]. In our hands, the use of liver as the reference standard performed better than IVC, confirming the literature data reported in naïve giant cell arteritis patients [29].

Some limitations should be acknowledged. First, this was a retrospective study in which clinical diagnosis was used as the reference standard. Images at the time of PET/CT were interpreted by different physicians with a diverse expertise in the field of inflammation, and medical reports might have impacted on the clinical diagnosis. Indeed, reader 1′s sensitivity was lower than that generally reported in the literature [10,11,20]. However, this was expected, since for the analysis, we did not use medical reports, but we visually re-assessed all images, rigorously applying the diagnostic criteria recommended by the current guidelines [11]. Accordingly, only patients presenting a vascular uptake higher than the liver was scored as positive. On the other hand, the final clinical diagnosis was realistically influenced by a positive (or negative) medical report, since, as shown by the literature, the incorporation of [^18^F]FDG PET/CT significantly impacted the patients’ LVV management [23]. Bearing this in mind, it is possible that the performance of reader 1 as well as that of reader 2 could have been different if only biopsy was considered as a valid final diagnosis, or if they performed image analysis at the time of diagnosis. Second, as already indicated, the majority of our patients with vasculitis were taking corticosteroid therapy, which intrinsically reduces the diagnostic performance of [^18^F]FDG PET/CT if lasting for more than 3 days or even at low-dosage levels. Third, the retrospective nature of the study prevented the possibility of evaluating other conditions or comorbidities as founders.

## 5. Conclusions

Our data confirmed that the specific imaging reading experience definitely affected the diagnostic performance of [^18^F]FDG PET/CT in diagnosing vasculitis, although the use of proper and specific criteria reduced the differences between the readers. Age, BMI, steroid use, and scanner quality influenced the visual image analysis, while other factors related to image acquisition such as glycemia and scanning time seemed to have less of an impact. Moreover, we confirmed the limited power of semi-quantitative analysis in LVV clinical practice.

## Figures and Tables

**Figure 1 diagnostics-12-02717-f001:**
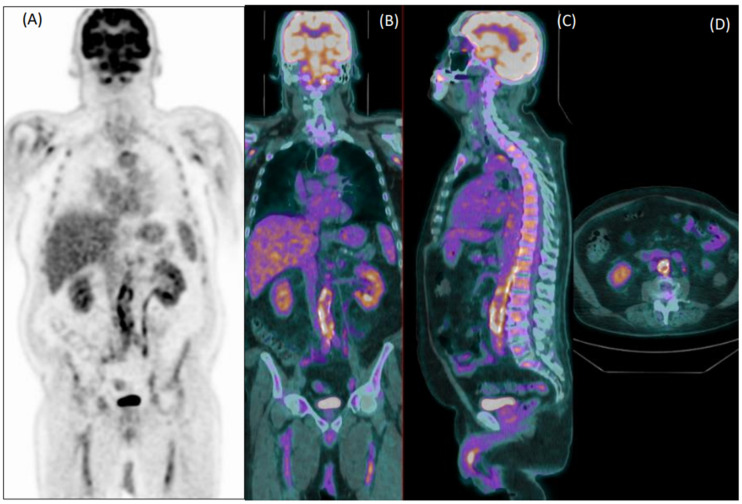
Coronal [^18^F]FDG PET (**A**) and fused PET/CT coronal (**B**), sagittal (**C**) and axial (**D**) images of an 80-year-old male patient with fever of unknown origin. Images show diffuse increased uptake in the abdominal aorta, common iliac, and femoral arteries. Both readers evaluated this exam as positive for vasculitis, which was confirmed during clinical follow-up.

**Figure 2 diagnostics-12-02717-f002:**
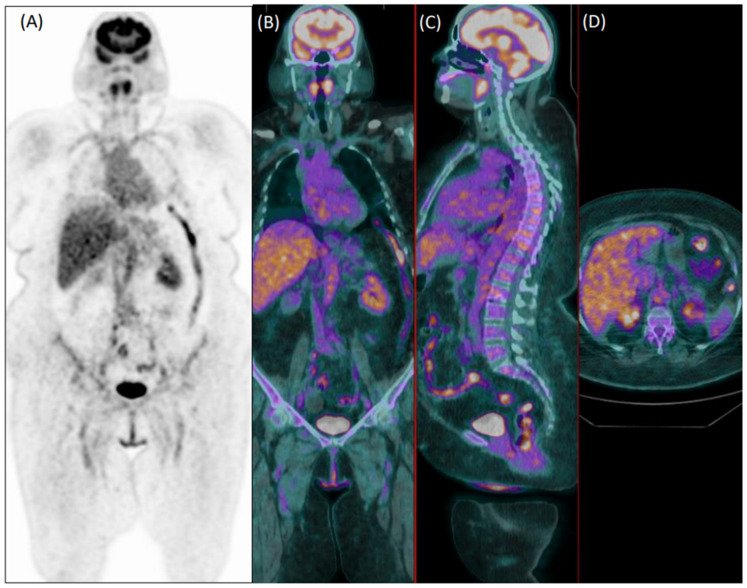
Coronal [^18^F]FDG PET (**A**) and fused PET/CT coronal (**B**), sagittal (**C**), and axial (**D**) images of a 75-year-old female patient with diffuse articular pain and suspicion of vasculitis, not further confirmed during clinical follow-up. Images were evaluated as positive for increased uptake in the aorta, and both subclavian and iliac arteries by the experienced reader 1. Reader 2 evaluated the exam as negative.

**Figure 3 diagnostics-12-02717-f003:**
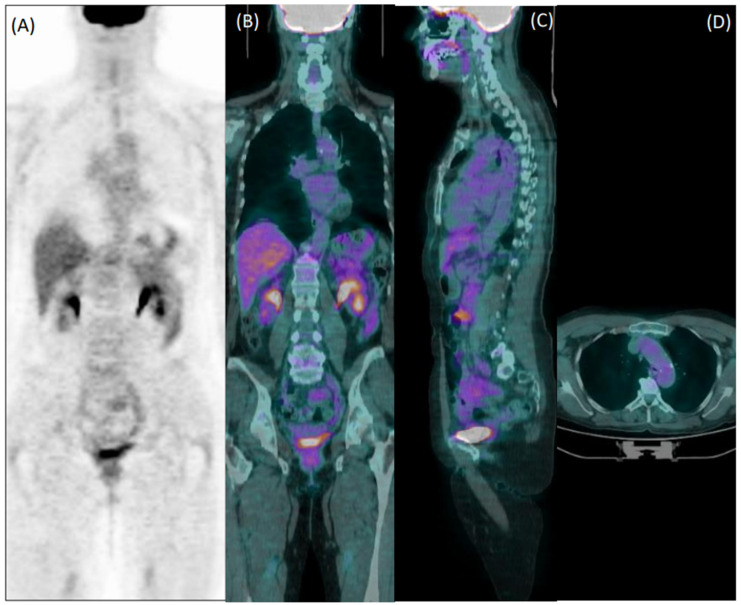
Coronal [^18^F]FDG PET (**A**) and fused PET/CT coronal (**B**), sagittal (**C**), and axial (**D**) images of a 60-year-old female patient with giant cell arteritis, who performed the exam in the course of corticosteroid therapy. Both readers evaluated images as negative for arteritis.

**Figure 4 diagnostics-12-02717-f004:**
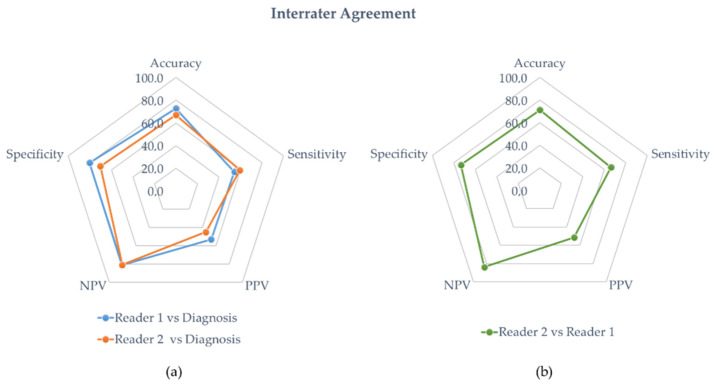
(**a**) Interrater agreement with diagnosis. The blue and orange line represent the diagnostic performances of the experienced physician (reader 1) and the student (reader 2), respectively. (**b**) Interrater agreement of the student’s diagnostic performances against the experienced physician (green line).

**Table 1 diagnostics-12-02717-t001:** Interrater agreement classification.

Value	Agreement
<0.0	Poor
0.00–0.20	Slight
0.21–0.40	Fair
0.41–0.60	Moderate
0.61–0.80	Substantial
0.81–1.00	Almost perfect

**Table 2 diagnostics-12-02717-t002:** The patients’ characteristics.

Characteristics	n (%)	Mean ± SD
Total patients	279	
Males	100 (36)	
Females	179 (64)	
Age (years)		65 ± 15
≤65	107 (38)	
>65	172 (62)	
BMI		26.05 ± 5.32
Normal (≤25)	141 (51)	
Overweight (>25)	138 (49)	
Glycemia (mg/dL)		102 ± 22
Normal ≤ 126	240 (86)	
Elevated > 126	39 (14)	
Corticosteroids		
Yes	150 (54)	
No	129 (46)	
[^18^F]FDG activity (MBq)		355 ± 18
Interval injection-imaging (minutes)		63 ± 8
EARL timing		
<55 (min 50)	36 (13)	
55–75	221 (79)	
75–112	22 (8)	
Timing cohort		
Early (<60 (min 50))	108 (39)	
Timely (60–65)	79 (28)	
Delayed (>65 (max 112))	92 (33)	
Scanner		
Siemens Biograph LS 6 (P1)	95 (34)	
GE Discovery 690 (P2)	184 (66)	

[^18^F]FDG: [^18^F]-fluorodeoxyglucose; BMI: body mass index; EARL: European Association of Nuclear Medicine Research Ltd.; P1: PET 1; P2: PET 2; SD: standard deviation.

**Table 3 diagnostics-12-02717-t003:** Diagnostic performances of reader 1 detailing the values of sensitivity, specificity, accuracy, PPN, NPV, and Cohen’s Kappa overall and in each subset of patients.

Population	Age		BMI		Steroids		Glycaemia		Interval Injection-Acquisition			PET Scanner	
	**≤65**	**>65**	**Normal**	**Over**	**Yes**	**No**	**≤126**	**>126**	**<60**	**60–65**	**>65**	**P1**	**P2**
Sensitivity	60.7	50.9	65.1	42.1	54.3	54.5	54.4	53.8	56.7	54.5	51.7	34.8	62.1
Specificity	82.3	79.0	82.7	78.0	82.5	78.8	80.8	76.9	79.5	80.7	81.0	88.9	75.4
Accuracy	76.6	70.3	77.3	68.1	69.3	76.7	73.3	69.2	73.1	73.4	71.7	75.8	71.2
PPV	54.8	51.9	62.2	42.1	73.1	19.4	52.9	53.8	51.5	52.2	55.6	50.0	53.7
NPV	85.5	78.3	84.4	78.0	67.3	94.9	81.8	76.9	82.7	82.1	78.5	81.0	81.2
**Cohen’s Kappa coefficient**													
Kappa	0.42	0.30	0.47	0.20	0.37	0.18	0.35	0.31	0.35	0.35	0.33	0.26	0.36

BMI: body mass index; NPV: negative predictive value; P1: PET 1; P2: PET 2; PET: positron emission tomography; PPV: positive predictive value.

**Table 4 diagnostics-12-02717-t004:** Diagnostic performances of reader 2 detailing the values of sensitivity, specificity, accuracy, PPN, NPV, and Cohen’s Kappa overall and in each subset of patients.

Population	Age		BMI		Steroids		Glycaemia		Interval Injection-Acquisition			PET Scanner	
	**≤65**	**>65**	**Normal**	**Over**	**Yes**	**No**	**≤126**	**>126**	**<60**	**60–65**	**>65**	**P1**	**P2**
Sensitivity	71.4	52.8	67.4	50.0	58.6	63.6	57.4	69.2	66.7	50.0	58.6	26.1	72.4
Specificity	67.1	72.3	62.2	78.0	75.0	66.9	68.0	84.6	74.4	70.2	65.1	93.1	57.1
Accuracy	68.2	66.3	63.8	70.3	67.3	66.7	65.0	79.5	72.2	64.6	63.0	76.8	62.0
PPV	43.5	45.9	43.9	46.3	67.2	15.2	41.5	69.2	50.0	39.3	43.6	54.5	43.8
NPV	86.9	77.5	81.3	80.4	67.4	95.2	80.1	84.6	85.3	78.4	77.4	79.8	81.8
**Cohen’s Kappa coefficient**													
Kappa	0.32	0.24	0.26	0.27	0.34	0.13	0.23	0.54	0.37	0.19	0.22	0.23	0.25

BMI: body mass index; NPV: negative predictive value; P1: PET 1; P2: PET 2; PET: positron emission tomography; PPV: positive predictive value.

## Data Availability

The manuscript represents valid work, and neither this manuscript nor one with substantially similar content under the same authorship has been published or is being considered for publication elsewhere. Arturo Chiti had full access to all of the data in the study and takes responsibility for the data integrity and the accuracy of the data analysis. Raw data are available on specific request to the corresponding author (10.5281/zenodo.7293127).

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
