# Peer review of "[18F]FDG PET/CT in Large Vessel Vasculitis: The Impact of Expertise and Confounders on Image Analysis"

_diagnostics, 2022, doi:10.3390/diagnostics12112717_

Round 1
Reviewer 1 Report
Specific comments/questions:
1) A main premise for the results of this study is that the reference standard is valid. Thus, a serious concern regarding the scientific soundness of this manuscript is the lack of description of the factors determining the final diagnosis of LVV. A thorough description in “Materials and Methods” is essential. In the present text, it is not possible to assess the strenght of this final diagnosis/gold standard. For instance, is the PET result included in this assessment? If so, a risk of confounding is present. In addition, how was results of temporal artery biopsies and ultrasound examinations weighed? How were ambiguous cases handled? Expert consensus?
2) The primary aim is unclear. Is it to assess differences in diagnostic performance between a specialist in nuclear medicine and a medical student or is it to determine the diagnostic performance of FDG PET in patients suspected of LVV including semi-quantitative PET measures?
3) Were extracranial arteries assessed? A patient may have giant cell arteritis without large vessel involvement – particularly in the vertebral arteries.
4) Most of the population received treatment with corticosteroids at the time of PET scanning; however, details about dosage and duration are lacking missing an important opportunity to do subgroup analyses, which would be very relevant as this is a common issue in this patient category.
5) It is not clear if only the EARL reconstruction was used, or if different proprietary reconstruction methods were used for the 2 different PET scanners.
6) It is not stated how many patients were identified in the initial search. This is important in the assessment of confounding. Hypothetically, clinically ambiguous patients are referred to FDG PET, while the easy cases are not. This may profoundly affect the diagnostic performance of PET.
Author Response
1) A main premise for the results of this study is that the reference standard is valid. Thus, a serious concern regarding the scientific soundness of this manuscript is the lack of description of the factors determining the final diagnosis of LVV. A thorough description in “Materials and Methods” is essential. In the present text, it is not possible to assess the strenght of this final diagnosis/gold standard. For instance, is the PET result included in this assessment? If so, a risk of confounding is present. In addition, how was results of temporal artery biopsies and ultrasound examinations weighed? How were ambiguous cases handled? Expert consensus?
We thank the reviewer for this comment which allowed to clarify this crucial issue. In our cohort, only a minority of patients with specific symptoms underwent temporal artery biopsy, since temporal involvement may not be present in LVV. Clinicians based the suspicious of LVV on clinical presentation and laboratory tests. Ultrasound in this specific population was not diriment as for it was not conclusive for LVV (see also answer n. 6), and PET/CT might have supported (or not) the final diagnosis of LVV as already stated in the discussion section regarding limitation of the study. As pointed out by the reviewer, taking all together these aspects may have affect reference standard. On the other hand, this bias is common to retrospective studies on PET/CT in LVV, since clinical diagnosis might consider positive or negative PET/CT for the final conclusion. More complex and inconclusive cases are in our Institution routinary handled by interdisciplinary discussion, and final diagnosis established accordingly. As requested, we further detailed this information in the text.
2) The primary aim is unclear. Is it to assess differences in diagnostic performance between a specialist in nuclear medicine and a medical student or is it to determine the diagnostic performance of FDG PET in patients suspected of LVV including semi-quantitative PET measures?
We realized that the primary aim was not clearly defined, therefore we changed the abstract and main text accordingly to make this point clearer.
3) Were extracranial arteries assessed? A patient may have giant cell arteritis without large vessel involvement – particularly in the vertebral arteries.
All vascular districts including extracranial involvement (e.g. temporal/vertebral arteries) were evaluated qualitatively by visual assessment, while as specified in the 2.3 section region of interests for the semi-quantitative analysis were drawn on 14 different vessels of interest including carotid, subclavian, axillary, iliac and femoral arteries bilaterally and aorta (ascending, arch, descending, and abdominal). For the sake of clarity we better specified this point in the text.
4) Most of the population received treatment with corticosteroids at the time of PET scanning; however, details about dosage and duration are lacking missing an important opportunity to do subgroup analyses, which would be very relevant as this is a common issue in this patient category.
We fully agree with the reviewer. Unfortunately, as mentioned in the text we were not able to consider the impact of duration and dosage of steroids treatment since these data were not retrospectively available for all patients. We further specified this limitation.
5) It is not clear if only the EARL reconstruction was used, or if different proprietary reconstruction methods were used for the 2 different PET scanners.
We thank the reviewer for this comment; we specified this point in the text.
6) It is not stated how many patients were identified in the initial search. This is important in the assessment of confounding. Hypothetically, clinically ambiguous patients are referred to FDG PET, while the easy cases are not. This may profoundly affect the diagnostic performance of PET.
The number of patients initially selected by using key words within medical reports was 432, as described in the 2.1 section. This selection was restricted to patients referred for PET/CT in the suspicious of vasculitis. Obliviously, this approach to retrospectively screen patients might introduce a bias since patients with a definite diagnosis of LVV based on clinics were not considered for PET/CT, and therefore not included in this study.
Reviewer 2 Report
Reviewer comments
18F-FDG PET-CT in large vessel vasculitis: the impact of expertise and confounders on image analysis
The topic is timely and interesting. However, some points need to be addressed.
|
Line |
Manuscript |
Comments |
||
|
13 |
This study aims 13 to evaluate the diagnostic accuracy of [18F]-fluorodeoxyglucose (FDG) |
The aim of the study is usually written in past tense The aim of the study needs to be modified to cope with the title |
||
|
19 |
Results |
The results usually describe the data in numbers and p value |
||
|
20 |
better compared to the less experienced reader |
Not cope with the aim of the study |
||
|
22 |
The diagnostic performance of both readers did not improve when semi-quantitative param-22 eters were added to the analysis |
Not cope with the aim of the study |
||
|
23 |
The results of our study underline the importance of 23 the experience in image interpretation for optimal diagnostic performance of [18F]-FDG PET/CT in 24 vasculitis. We also confirmed the significance of the use of specific criteria for image acquisition 25 and interpretation. |
The conclusion does not reflect the aim of the study |
||
|
34 |
with the latter manifesting an increased risk in women |
English editing is needed |
||
|
39 |
GCA appears with the abrupt onset |
English editing is mandatory |
||
|
50 |
specificity also at the earlier phases |
Inappropriate structure of many sentences |
||
|
51 |
the EANM, SNMMI |
What are these abbreviations stand for ? |
||
|
57-60 |
|
The last part of the introduction section should include the research gap and the aim of the study The authors added the summary of the materials and methods at the last paragraph of the introduction which is not appropriate choice for that paragraph |
||
|
65 |
IRCCS |
What does this abbreviation stand for |
||
|
71 |
We collected from the 71 electronic clinical medical records demographic patient characteristics |
Inappropriate structure of the sentences |
||
|
78 |
Elderly patients (> 65 years) accounted for 172 out of 279 cases (62%). BMI was 78 normal (≤25) in 141/279 |
The data and statistics of patients are usually written in the results section |
||
|
213 |
|
Any abbreviations in the table should be determined at the footnote |
||
|
216 |
|
What is the value of adding the performance of reader 1 and then reader 2 This is not the aim of the study |
||
|
217 |
comparing reader ability of an experienced nuclear medicine physician 247 to a medical student. |
The authors changed the aim of the study mentioned at the abstract |
Author Response
|
Line |
Manuscript |
Comments |
||
|
13 |
This study aims 13 to evaluate the diagnostic accuracy of [18F]-fluorodeoxyglucose (FDG) |
The aim of the study is usually written in past tense The aim of the study needs to be modified to cope with the title |
||
|
|
|
We amended the text according to the Reviewer’s comment |
||
|
19 |
Results |
The results usually describe the data in numbers and p value |
||
|
|
|
The numeric values were added to the results session of the abstract, as suggested. |
||
|
20 |
better compared to the less experienced reader |
Not cope with the aim of the study |
||
|
22 |
The diagnostic performance of both readers did not improve when semi-quantitative param-22 eters were added to the analysis |
Not cope with the aim of the study |
||
|
23 |
The results of our study underline the importance of 23 the experience in image interpretation for optimal diagnostic performance of [18F]-FDG PET/CT in 24 vasculitis. We also confirmed the significance of the use of specific criteria for image acquisition 25 and interpretation. |
The conclusion does not reflect the aim of the study |
||
|
|
|
With regard to the preceding three points, we changed the abstract and main text to make the primary aim of the study clear. |
||
|
34 |
with the latter manifesting an increased risk in women |
English editing is needed |
||
|
|
|
We edited this text |
||
|
39 |
GCA appears with the abrupt onset |
English editing is mandatory |
||
|
|
|
This sentence has been edited as requested |
||
|
50 |
specificity also at the earlier phases |
Inappropriate structure of many sentences |
||
|
|
|
We have substantially amended the periods in this section of the manuscript |
||
|
51 |
the EANM, SNMMI |
What are these abbreviations stand for ? |
||
|
|
|
These abbreviations have been spelt out |
||
|
57-60 |
|
The last part of the introduction section should include the research gap and the aim of the study The authors added the summary of the materials and methods at the last paragraph of the introduction which is not appropriate choice for that paragraph |
||
|
|
|
The research gap and the aim of the study have been better defined in the revised version of the manuscript. Also, the last part of the introduction has been modified. |
||
|
65 |
IRCCS |
What does this abbreviation stand for |
||
|
|
|
This abbreviation is part of the name of the Hospital, so we left it as in the text. |
||
|
71 |
We collected from the 71 electronic clinical medical records demographic patient characteristics |
Inappropriate structure of the sentences |
||
|
|
|
This sentence has been rephrased for sake of clarity |
||
|
78 |
Elderly patients (> 65 years) accounted for 172 out of 279 cases (62%). BMI was 78 normal (≤25) in 141/279 |
The data and statistics of patients are usually written in the results section |
||
|
|
|
We took the Reviewer’s point and moved the description of patients’ characteristics and the corresponding table to the Results section. |
||
|
213 |
|
Any abbreviations in the table should be determined at the footnote |
||
|
|
|
We appreciate the Reviewer’s point and point out that this abbreviations are defined in the Materials and Methods section (paragraph 2.2) as well as in Tables. |
||
|
216 |
|
What is the value of adding the performance of reader 1 and then reader 2 This is not the aim of the study |
||
|
|
|
Following upon the Reviewer’s comments, we have better detailed the aim of the study. In this regard, we highlight here that visual assessment of PET images in LVV is still largely operator-dependent, a point that was accounted for in the study design. |
||
|
217 |
comparing reader ability of an experienced nuclear medicine physician 247 to a medical student. |
The authors changed the aim of the study mentioned at the abstract |
||
|
|
|
Please see our response to the point raised above. |
||
|
|
|
|
Round 2
Reviewer 2 Report
The manuscript was improved greatly